# The Effect of a Six-Week Nordic Walking Training Cycle on Oxidative Damage of Macromolecules and Iron Metabolism in Older Patients with Multiple Myeloma in Remission—Randomized Clinical Trial

**DOI:** 10.3390/ijms242015358

**Published:** 2023-10-19

**Authors:** Olga Czerwińska-Ledwig, Artur Jurczyszyn, Anna Piotrowska, Wanda Pilch, Jędrzej Antosiewicz, Małgorzata Żychowska

**Affiliations:** 1Department of Chemistry and Biochemistry, Institute of Basics Sciences, Faculty of Physiotherapy, University of Physical Education, 31-571 Kraków, Poland; olga.czerwinska@awf.krakow.pl (O.C.-L.); anna.piotrowska@awf.krakow.pl (A.P.); wanda.pilch@awf.krakow.pl (W.P.); 2Plasma Cell Dyscrasia Center, Department of Hematology, Faculty of Medicine, Jagiellonian University Medical College, 31-501 Krakow, Poland; mmjurczy@cyf-kr.edu.pl; 3Department of Bioenergetics and Exercise Physiology, Medical University of Gdansk, 80-210 Gdansk, Poland; 4Department of Biological Foundations of Physical Culture, Faculty of Health Science and Physical Culture, Kazimierz Wielki University, 85-091 Bydgoszcz, Poland; mz0511@ukw.edu.pl

**Keywords:** multiple myeloma, physical activity, walking with poles, iron metabolism, oxidative damage

## Abstract

Multiple myeloma (MM) is an incurable hematologic malignancy originating from clonal plasma cell proliferation within the bone marrow, predominantly affecting older individuals. While anemia serves as a diagnostic criterion for MM, it often ameliorates upon achieving disease remission. Iron metabolism parameters have emerged as potential prognostic indicators in MM. Notably, physical exercise has been established to influence iron metabolism. This study aimed to assess alterations in serum iron, ferritin, and transferrin concentrations, as well as leukocyte gene expression, in MM patients undergoing a six-week cycle of Nordic walking training. Thirty patients divided into an exercise group (NW, n = 15, mean age 63.1 ± 8.4 years) and a control group (CG, n = 15, mean age: 63.5 ± 3.6 years) completed the study protocol. Blood samples were collected at baseline, after three and six weeks of training, and after nine weeks. Serum ferritin, transferrin, and iron concentrations were measured, along with the leukocyte expression of genes. Additionally, serum oxidative damage marker levels were determined. Following the Nordic walking training cycle, a declining trend in serum ferritin concentrations was observed. Intracellular mRNA levels of genes associated with iron metabolism were positively influenced by the training regimen, indicating the potential impact of this physical activity on gene expression and ferritin concentrations. Although positive trends were noted, extended training periods might be requisite for significant changes. To conclude, moderate-intensity exercise induces favorable shifts in the analyzed parameters among MM patients, potentially influencing disease progression. Consequently, Nordic walking training is a safe recommendation for MM patients, though sustained training beyond six weeks could be necessary for notable effects on iron metabolism factors.

## 1. Introduction

Multiple myeloma (MM) is a hematologic malignancy originating from plasma cells found in the bone marrow. This disease accounts for approximately 10 to 15% of all diagnosed hematologic malignancies and approximately 1–2% of all cancer cases [1]. The median survival in relation to the stage of advancement at the time of diagnosis ranges from 29 to 62 months. MM primarily affects older individuals (median age at diagnosis: approximately 70 years) and occurs more frequently in males than females [2]. In patients, the neoplastic transformation results in the development of clonal plasma cell clones in the bone marrow, which typically produce a monoclonal protein detectable in the serum and urine. The presence of this protein, along with the activity of myeloma cells, leads to a series of organ-related consequences used for diagnosing MM [2].

Iron homeostasis is of great importance in patients with MM, as they often experience anemia (approximately 75% of patients), which is one of the diagnostic criteria for this disease [2,3]. It is associated with a reduction in red blood cell production which is related with infiltration of the bone marrow by MM cells, and may also be a side effect of treatment [4]. The serum parameters related with iron metabolism, such as iron or ferritin concentrations as well as intracellular expression of genes such as *PCBP1* (poly(rC) binding protein 1), *PCBP2* (poly(rC) binding protein 2), *FTH1* (ferritin heavy chain 1), *FTL* (ferritin light chain), and *TFRC* (transferrin receptor 1), which are involved in iron homeostasis, are induced by stress stimuli, among which are physical exercise. Oxidative stress and inflammation parameters are usually increased after exercise. This effect is particularly noticeable after performing intense physical exertion, such as running a marathon [5]. Data on the effect on reactive oxygen species (ROS) generation as a result of moderate-intensity exercise are inconclusive (as shown in extensive reviews by Thirupathi et al. [6] and Finaud et al. [7]. Some studies indicate that this type of exercise does not cause an increase in the production of ROS, but other authors point out that even low-intensity exercise can cause their production depending on time and intensity and inefficient antioxidant mechanisms [6]. However, it is worth mentioning that adaptive changes to prolonged exercise, especially on mRNA levels, are not well documented. Intracellular changes in blood cells among genes encoding *PCBP1*, *PCBP2*, *FTL*, and *FTH* lead to understanding changes in regulation of iron economy in white cells and could be independent of plasma indicators in healthy and ill people [8].

Oxidative stress is one of the main factors which influences iron metabolism and the expression of genes associated with it. The level of oxidative stress and oxidative damage increases with age [9] and in the course many diseases including MM. Damage caused by free radicals and ROS occurs in all macromolecules, and their markers include compounds such as 3-nitrotyrosine (3-NT) which is formed as a result of ROS action on protein molecules and trans-4-hydroxy-2-nonenal (4-HNE) which along with malondialdehyde (MDA) is a major highly mutagenic product of lipid peroxidation [10,11]. In the assessment of oxidative damage, another commonly used compound is 8-iso-prostaglandin F2α (8-iso-PGF2α), which is formed through the oxidation of arachidonic acid. Additionally, 8-OH-deoxyguanosine (8-OHdG) serves as a marker for DNA exposure to free radicals [12].

Free radical-induced damage also plays a significant role in MM. Patients with MM have been shown to exhibit high levels of oxidative stress compared to healthy individuals. This is associated with decreased activity of antioxidant enzymes in these patients and increased levels of lipid peroxidation products, protein oxidation, and DNA damage [13]. Therefore, moderate physical activity may cause an increase in antioxidant capacity, observed as an adaptive response to exercise by many authors [14,15,16] and could be well tolerated by patients with MM.

Walking with poles, also known as Nordic walking, is a safe form of physical activity recommended for both older individuals and patients with hematologic malignancies. In patients with MM in remission, it has been shown to have a beneficial effect on disease parameters [17] as well as calcium-phosphate metabolism [18]. However, there are no studies evaluating the impact of this form of activity on the blood biochemical parameters related to oxidative damage of macromolecules and iron metabolism in patients with MM. The well-documented beneficial effects of physical exercise in the form of Nordic walking on the mentioned parameters have prompted the authors of the project to undertake this research topic.

The aim of this study was to assess the impact of a six-week cycle of Nordic walking training on iron metabolism parameters and the expression of related genes in patients with MM. Additionally, this study aimed to evaluate the effects of oxidative stress by measuring the serum concentrations of products of oxidative modifications of macromolecules.

## 2. Results

### 2.1. Study Group Characteristics

The age of the study group performing Nordic walking trainings cycles (NW, n = 15) was 63.1 ± 8.4 years, and the time since diagnosis of MM was 39.8 ± 13.8 months. In the control group (CG, n = 15), these parameters were 63.5 ± 3.6 years and 36.3 ± 16.6 months, respectively. The average BMI of the NW group was 29.12 ± 4.01 kg∙m^−2^ and the average CG was 28.77 ± 4.18 kg∙m^−2^. No statistical differences between the selected groups were indicated. All patients enrolled in the study were in remission stage of MM and had undergone the autologous hematopoietic stem cells transplantation (ASCT) procedure at least six months before the beginning of the study and achieved remission. Among NW patients, IgG myeloma was diagnosed in eight subjects (IgG kappa in five, lambda in three), IgA—in six patients (kappa in four, lambda in two), and in one subject light chains myeloma was detected. In CG, four patients each were diagnosed with IgG kappa and IgG lambda myeloma. IgA kappa MM type was diagnosed in five participants and IgA lambda in two.

### 2.2. Parameters of Blood Cells Count

The white blood cells count, red blood cells count, and hemoglobin concentrations in all time points are shown in Table 1. There were no statistical differences between the groups and statistically significant differences were observed only in the NW group in WBC and RBC counts.

### 2.3. Parameters of Iron Metabolism in Serum

Table 2 presents the concentrations of iron metabolism markers in the blood for both groups participating in the study in all four time points. All the results showed that the concentrations of tested parameters were within reference ranges.

Significant differences between groups in iron, transferrin, and transferrin saturation (TfS) (Cohen’s d—medium to large effect) were observed, resulting from the lower concentrations obtained in the control group. Ferritin concentration was higher in the control group and no correlation between iron and ferritin was found in both groups. The first 3 weeks of training caused a slight decrease in iron, transferrin, and TfS, but not ferritin concentration. During the next 3 weeks of training sessions, further decreases in iron concentration and TfS were recorded, while transferrin remained unchanged. A slight increase in ferritin serum concentration was observed at the same time. However, 6 weeks of NW training caused statistically significant changes only for iron concentration. In the control group, iron and TfS slightly increased, but ferritin slightly decreased, especially after 6 weeks from the start of the experiment.

Two-way ANOVA showed significant differences in time, row factor, and interaction between groups. It was observed that 3 weeks after the end of the experiment, iron, ferritin, and transferrin concentrations were slightly lower than at baseline only in the NW group. Cohen’s d showed that 6 weeks of applied training caused lower differences between groups for all parameters compared to baseline; however, this effect did not last up to 3 weeks after intervention.

### 2.4. Macromolecules Oxidative Damage Parameters

The results obtained in both groups in all four time points for macromolecule damage indicators are presented in Figure 1A–D.


**Baseline**


No differences in 3NT (Figure 1A), 4HNE (Figure 1B), and 8dOHG (Figure 1C) concentration were observed at baseline between the CG and NW groups, and significantly higher levels were found for 8-iso-PGF2a (Figure 1D) concentration in CG (dCohen = 2.08—strong).


**Effects of NW training**


It was observed that 3-NT (Figure 1A) and 4HNE (Figure 1B) concentration slightly decreased in the NW group in all time points compared to baseline. However, this change was significant only for 4HNE after 3 weeks of training, after which it also significantly increased during the next six weeks of training.

Changes in 8-dHG (Figure 1C) and 8-isoPGFa2 (Figure 1D) during intervention showed a significant increase in this parameter compared to baseline after 3 and 6 weeks of training and remained after the end of the training period. No effect of training was observed in 8-isoPGFa2 concentration.


**Differences between groups**


A significant difference was observed after 3 weeks of training in 3-NT concentration (Figure 1A); however, it was lowered in this parameter in all time points except baseline in the NW group. A 2-way ANOVA showed significant differences for interaction, row factor, and time. Cohen d analysis showed no differences at baseline (0.03), strong after 3 weeks of training (0.84), and moderate after 6 weeks of training (0.64) and during the next weeks of training (0.72).

The 4HNE (Figure 1B) concentration was lower in the NW group in all time points; however, significant changes were observed after 3 and 6 weeks of Nordic walking training. A 2-way Anova confirmed significant differences for interaction, row factor, and time. Cohen d analysis showed that higher effect was observed after 3 weeks of training and at the end of the experiment (effect size in each time point was 0.74; 1.19; 0.78; 1.07).

Differences between groups in 8dOHG (Figure 1C) relied on lower (not significant) concentration in CG in all time points except baseline. In both groups starting from 3 weeks after Nordic walking training, the tendency to slight decreases was similar. A 2-way ANOVA showed significant differences only for subject. Cohen d analysis showed moderate and strong (after 6 weeks of training) effect size which was 0.61; 0.48; 0.92; 0.7.

Significant differences between baseline and after 3 weeks of the end of the experiment concerned 8-isoPGFa2 (Figure 1D) which was lower in the NW group. The concentration of this indicator was high in CG at baseline and decreased in the next two time points. This change makes interpretation of the results difficult. A 2-way ANOVA showed significant differences only for time. Cohen d effect size was as follows: 2.08 (strong) at baseline; 0.16 (small) after 3 weeks of training; 0.34 (small) after 6 weeks of training; and 1.14 (strong) after 3 weeks of the end of training.

The reference values for these oxidative stress parameters are not clearly determined, but based on literature data, the following ranges found in young healthy individuals can be established: 8-OHdG—16.95 ± 10.66 ng/mL [20], 8-isoPGF2a—42 pg/mL (24–70 pg/mL) [21], 3-NT—<22 nmol/L [22], 4-HNE—0.65 ± 0.39 μM (101 ± 61 pg/mL) [23].

### 2.5. Gene Expression in Leukocytes

Changes in intracellular *FTH*, *FTL*, *TFRC*, *PCBP1*, and *PCBP2* genes expression in all four times in both group points are shown in Figure 2A–E.


**Baseline**


No significant changes at baseline in genes mRNA were observed between the tested groups; however, *FTH* and *FTL* mRNA were slightly higher in the NW group (Cohen d 0.37—moderate for *FTH* mRNA and 0.53 for *FTL* mRNA, Figure 2A,B).


**Effects of NW training**


*FTH* and *FTL* mRNA consequently decreased after 3 and 6 weeks of training; however, this effect did not last 3 weeks after exercise (Figure 2A,B). For *FTL* mRNA, a significant change was observed after 6 weeks of training. Tendency to decreases was noted also for *TFRC* mRNA (Figure 2C). Changes in this gene expression were significant after 3 weeks of training and remained lower after 6 weeks. This effect was visible only in the training period, whereby 3 weeks after training this expression increased even above the baseline value. *PCBP1* and *PCBP2* mRNA had a tendency to increase after 6 weeks NW training and after 3 weeks of the end of the experiment a significant increase could be observed (Figure 2D,E).


**Differences between groups**


Significant differences in genes expression were observed for all genes excluding *FTH* mRNA (Figure 2A); however, these differences were noted in different time points: for *FTL* mRNA 6 weeks after training (Figure 2B), for *TFRC* mRNA 3 weeks after training (and this change was clear also 6 weeks after training, Figure 2C), for *PCBP1* mRNA 6 weeks after training and 3 weeks after the end of the experiment (Figure 2D), for *PCBP2* 3 weeks after the end of the experiment (Figure 2E). A 2-way ANOVA confirmed significant differences between groups for time, row factor, and interaction for all tested genes. Cohen d analysis showed small, moderate, and strong effect sizes for tested genes. In detail: for *FTH* mRNA: 0.37; 0.16; 0.35; 0.25; for *FTL* mRNA: 0.53; 0.14; 0.31; 0.02; for *TFRC* mRNA: 0; 0.74; 0.56; 0.17; for P*CBP1* mRNA: 0.03; 0.18; 0.96; 2.63; and for *PCBP2* mRNA: 0.36; 0.05; 0.79; 1.52.

## 3. Discussion

The main aim of our study was to answer how 3 and 6 weeks of Nordic walking training influence iron metabolism in plasma and within white blood cells and on selected markers of oxidative stress. The obtained results were compared with a non-exercising control group also consisting of patients in disease remission.

### 3.1. Iron Metabolism in Multiple Myeloma

No significant deviations from laboratory norms were observed in the parameters of red blood cell morphology in the peripheral blood of MM patients participating in this study. This further confirms other clinical data indicating remission of the disease, as RBC parameters normalize in most patients at this stage [24]. Regarding serum ferritin concentration, Lodh et al. [13] demonstrated that newly diagnosed patients had statistically significantly higher concentrations of this protein compared to healthy individuals (285.8 ± 216.4 ng/mL versus 131.8 ± 30.1 ng/mL). In our own study, the concentrations of this protein were significantly lower than those of the patients participating in the cited study, which may have been explained by the fact that the patients were in MM remission. Zhang et al. demonstrated that serum ferritin concentrations change dynamically in different stages of this disease and can increase during relapse; therefore, the concentration of this protein can be used as a prognostic marker [25]. In our study, the mean ferritin concentrations in both groups did not exceed 150 ng/mL throughout the experiment, which indicates a favorable prognosis for the patients, as well as the absence of negative effects of the applied training intervention on the concentration of this protein.

Increased expression of *FTL* and *FTH* genes encoding ferritin chains, as well as elevated serum ferritin concentrations, have prognostic significance in another hematologic malignancy, acute myeloid leukemia. In a retrospective study by Bertoli et al., it was demonstrated in a group of 525 patients (mean age 59.4 years) that disease-free survival was significantly shorter in individuals with serum ferritin concentrations above 2100 μg/L, and overall survival was three times longer in patients with ferritin concentrations at diagnosis below 900 μg/L [26]. In the mentioned study, ferritin concentrations and the expression of *FTL* and *FTH* genes in leukemic cells also increased in patients undergoing chemotherapy, similar to other acute-phase proteins, including ferritin. In our study, comparable mRNA levels of the investigated genes were found, similar to those of healthy individuals in the study by Borkowska et al. [8]. In summary, important indicators related to the cellular profile of blood and iron metabolism did not deviate from the results obtained in healthy individuals, which can be considered as confirmation of disease remission.

### 3.2. Iron Metabolism and Nordic Walking

Nordic walking is an activity with well-documented beneficial effects on various biochemical parameters of blood, including iron metabolism [27,28,29]. In our study, the NW and CG groups (except for the sample taken after 6 weeks of training) differed significantly in terms of ferritin concentrations, but in both cases, the levels were within the physiological range. Kortas et al. demonstrated that serum ferritin concentration significantly decreased after 32 weeks of Nordic walking trainings—from 110.2 ± 82.9 ng/mL to 89.1 ± 66.7 ng/mL [27]. In our study, a decreasing trend in ferritin concentrations was observed 3 weeks after the completion of training in the NW group (follow up). It was found that the duration of 3 and 6 weeks of training was too short to induce significant changes in ferritin concentrations in patients. Intracellular changes in *FTL* and *FTH* mRNA levels showed a tendency towards decreased expression of these genes after 3 and further after 6 weeks of NW training (statistically significant for *FTL* mRNA), but this effect did not persist 3 weeks after the end of the experiment, which further encourages conducting studies with a longer duration of NW activity. This decrease was accompanied by an increase in WBC, which further confirms the observed trend.

The observed changes in serum Fe and gene expressions may be associated with a decrease in the labile iron pool (LIP), and therefore they should be considered beneficial, as free Fe ions are susceptible to oxidation and participate in the Fenton reaction, generating free radicals. An increase in LIP leads to an increase in ferritin levels in cells. According to the literature, *PCBP1* and *PCBP2* are iron chaperones that deliver iron to ferritin, the iron storage protein [30]. In our study, a significant increase in their mRNA levels 3 weeks after the completion of training (with a rising trend from 6 weeks of NW) was accompanied by an increase in *FTH* and *FTL* mRNA expression. Indirectly, this may indicate an increase in the iron pool [8].

High iron stores are associated with an increased risk of developing various non-communicable diseases, including cancer. Iron overload is also linked to a significant increase in the risk of infection in patients undergoing auto-HSCT procedures, which are often used in the treatment of MM [31]. In a study by Kortas et al. [28], a statistically significant decrease in serum iron concentrations from 99.36 ± 62.69 ng/mL to 81.43 ng/mL was observed after 12 weeks of NW training in 35 older women. Our findings are consistent with the results of Kortas et al. after 32 weeks and 12 weeks of training in older women. However, these changes were not statistically significant, which could be attributed to the relatively short duration of training in this study (6 weeks).

In our own study, no changes in serum transferrin concentrations, which is responsible for delivering iron to erythroblasts [30], were observed. Data regarding the impact of exercise on transferrin concentrations are not consistent and seem to be related to the intensity of exercise. Schumacher et al. demonstrated an increase in transferrin concentrations during laboratory treadmill tests at 70% VO2max intensity, while no such changes were observed after aerobic exercises on a cycle ergometer for four consecutive days [32]. Our results are consistent with the findings published by Schumacher et al. [32], who did not observe changes in transferrin concentrations after moderate-intensity cycle ergometer exercise, as well as Cichoń et al. [33] who did not find changes in transferrin concentrations immediately after an incremental treadmill running test or 3 h after its completion in women engaged in basketball training. Intracellular changes in *TFRC* mRNA were different from changes in serum. A clear trend of decreased expression of this gene was observed after 3 weeks of training, which persisted until the sample collection after 6 weeks, followed by an increase to initial values during the follow-up period, indicating a lasting effect (similar directional changes as *FTH* and *FTL*, although noted after 3 weeks of exercise).

### 3.3. Oxidative Stress in Multiple Myeloma

In patients with MM, there is an increase in oxidative stress levels, which is particularly evident during the treatment of the disease [34]. Ansari et al. demonstrated that the concentration of 8-OHdG is dependent on the stage of the disease [35]. The study also indicated that the levels of reactive nitrogen species are higher in newly diagnosed patients with MM compared to the healthy population [35]. Similar results were obtained in a study by Tandon et al. involving patients with MM and lymphoma [36]. Additionally, Sabuncuoğlu et al. found in a group of 16 patients that the concentration of markers of oxidative protein damage, such as 3-nitrotyrosine in red blood cells and carbonyl derivatives in serum, increased in patients undergoing HSCT transplantation procedures [37]. In our own study, the concentration of these oxidative stress markers was similar to the concentrations found in healthy individuals, which may indicate the favorable pro-oxidant-antioxidant status of the subjects in this study.

### 3.4. Oxidative Stress and Nordic Walking

Changes in 3-NT and 4-HNE concentrations showed that Nordic walking training induced a decrease in the concentration of these indicators (for 3-NT after 3 weeks of training, for 4-HNE after 3 and 6 weeks). Moreover, these changes were observed also 3 weeks after the training period (follow up). Changes in 8-OHdG and 8-isoPGFa2 were ambiguous, and no effect of exercise was observed for 8-OHdG, but there was a tendency to increase in 8-isoPGFa2. Similar results were obtained by Kortas et al. [38] in response to a 12-week Nordic walking program, examining other markers of protein (advanced oxidation protein products) and lipid peroxidation (MDA). However, we did not evaluate antioxidant capacity and this is a limitation of our study. Increases of some parameters of oxidative stress after exercises were reported in many studies, but these increases are accompanied by increases in antioxidants [39,40].

## 4. Materials and Methods

### 4.1. Study Group

The study group consisted of 30 patients with MM recruited among the patients of the Hematology Clinic at the Jagiellonian University Medical College in Krakow. The patients were randomly assigned to one of two groups: the Nordic walking training group (NW, n = 15, including ♂ = 7, ♀ = 8) and the control group, which has not been subjected to training intervention (CG, n = 15, including ♂ = 8, ♀ = 7). For random assignment to groups, a permuted block randomization procedure (ratio 1:1) was used. Each participant drew an opaque envelope containing an assignment for one of the groups. CG patients were asked not to change their physical activity habits during the project period. Patient flow diagram is shown in Figure 3.

Each patient received a daily supplementation of 2000 U of vitamin D3 and 1000–1500 mg of calcium carbonate. Additionally, in accordance with the prevailing guidelines of the International Myeloma Working Group [41], intravenous administration of 4 mg zoledronic acid was provided every 4 weeks.

Inclusion in the study was based on a detailed medical interview and meeting the inclusion criteria (Table 3). The participants were thoroughly informed about the study protocol and had the right to withdraw their consent at any stage of the project without consequences. The study received approval from the bioethics committee at the Regional Medical Chamber in Krakow (166/KBL/OIL/2018), and the entire project was registered in the ANZCTR clinical trial registry (trial ID: ACTRN12622000268741).

### 4.2. Study Protocol

Patients in the NW group underwent a moderate-intensity Nordic walking training cycle (60–70% of HRmax) for a period of 6 weeks. Venous blood samples were collected from all participants at baseline (I—before the start of the project), in the middle of the training cycle before the training session (II—3 weeks), after the completion of the project (III—6 weeks), and 3 weeks after the completion (IV—9 weeks, follow up). 

### 4.3. Methods

#### 4.3.1. Venous Blood Collection

Venous blood samples were collected from the participants by a qualified laboratory diagnostician using a vacuum system (BD Vacutainer). Each time, three tubes were collected: one with an anticoagulant (EDTA) and two containing a clot activator.

#### 4.3.2. Gene Expression Analysis

In detail, the methods and primers applied in this study were previously described by Grzybkowska et al. [5]. Leukocytes were isolated from the 2 mL of whole blood collected in the tube containing EDTA by erythrocyte lysis (red blood cell lysis buffer RBCL, A&A Biotechnology, Gdynia, Poland). Next, Fenozol was used (A&A Biotechnology, Gdynia, Poland) to release RNA from the cells. Further isolation was performed according to the Chomczynski and Sacchi method [42]. After assessing purity and quality of obtained RNA (Multiskan Sky Microplate Spectrophotometer, ThermoFisher Scientific, Warszawa, Poland), 1000 ng of RNA was subjected to reverse transcription (First Strand cDNA Synthesis Kit as per the manufacturer’s instructions, Roche, Warszawa, Poland), followed by real-time PCR (with use of polymerase Brillant III, Agilent, Poland). To determine the expression of the following genes: *FTH1* (ferritin heavy chain 1), *FTL* (ferritin light chain), *TFRC* (transferrin receptor), *PCBP1* (poly(rC)-binding protein 1), *PCBP2* (poly(rC)-binding protein 2), primers with the following 5′-3′ sequences were used (Genomed, Warsaw, Poland):


*FTH1*


Forward primer: *TCCTACGTTTACCTGTCCATG*

Reverse primer: *CTGCAGCTTCATCAGTTTCTC*


*FTL*


Forward primer: *GTCAATTTGTACCTGCAGGCC*

Reverse primer: *CTCGGCCAATTCGCGGAA*


*TFRC*


Forward primer: *TGCAGCAGTGAGTCTCTTCA*

Reverse primer: *AGGCCCATCTCCTTAACGAG*


*PCBP1*


Forward primer: *AGAGTCATGACCATTCCGTAC*

Reverse primer: *TCCTTGAATCGAGTAGGCATC*


*PCBP2*


Forward primer: *TCCAGCTCTCCGGTCATCTTT*

Reverse primer: *ACTGAATCCGGTGTTGCCATG*

*TUBB*—tubulin beta class 1 (used as the reference gene)

Forward primer: *TCCACGGCCTTGCTCTTGTTT*

Reverse primer: *GACATCAAGGCGCATGTGAAC*

The sequence of primers has been designed by the authors with the use of the Primer3 web specificity and has been screened in silico via the UCSC genome browser. The relative mRNA expression of *FTH1* (NM_002032), FTL (NM_000146), *TFRC* (NM_001128148), *PCBP1* (NM_006196), and *PCBP2* (NM_001128913) was calculated using qRT-PCR.

#### 4.3.3. Investigation of Serum Iron Metabolism Parameters

The routine parameters of iron metabolism in serum were analyzed in a medical diagnostic laboratory. The concentrations of ferritin and transferrin were determined using an immunoturbidimetric method on the Cobas 8000 analyzer (Roche, Indianapolis, IN, USA). The concentration of iron in the tested samples was measured using a colorimetric method on the Cobas 8000 analyzer (Roche, Indianapolis, IN, USA).

#### 4.3.4. Investigation of Oxidative Stress Parameters

The blood collected for biochemical analysis (in a tube containing coagulation activator) was centrifuged for 10 min at 4 °C and 2500 rpm. The resulting serum was pipetted into Eppendorf-type tubes and stored in a low-temperature freezer at −80 °C until further analysis. The assessment of oxidative damage parameters (concentrations of 8-OHdG, 8-iso-PGF2a, 4-HNE, and 3-NT) was performed using an immunoenzymatically method (ELISA test).

### 4.4. Nordic Walking Trainings

The training sessions were conducted outdoors by a qualified Nordic walking trainer during the spring and summer seasons. The intensity of the training was set at 60–70% of the individual’s maximum heart rate (HRmax), calculated using the Nes formula [43]. The intensity of the training was monitored with use of sport testers (Polar, Finland) in which data, including the minimal and maximal permissible heart rate levels, was input individually for each user. If the heart rate exceeded the permissible levels, the device emitted an audible signal, and the instructor adjusted the intensity of the exercises for the participant.

The health training program, lasting 6 weeks, consisted of 18 training sessions held 3 times a week in the morning, with each session lasting approximately 60 min. Each training session was divided into three parts: 5-min warm-up (low intensity), main part lasting 45 min (moderate intensity), 10 min of cool down exercises. During the main part, exercises with poles were performed, focusing on maintaining proper technique. The walking distance was progressively increased with each training session.

### 4.5. Statistical Analysis

For genes expression, the delta Tc method was applied [44]. Calculation of relative expression/TUBB was performed in EXEL 2022. Next, after transformation obtained data to linear values, the analysis of results was the same as other data. All obtained results were subjected to descriptive statistics analysis, including arithmetic mean, standard deviation (SD), and *p* values paired or unpaired (statistically significant values were considered under *p* < 0.05). The normality of the distribution was assessed using the Shapiro-Wilk test. For variables with a normal distribution, a two-way ANOVA analysis was performed, while for non-normally distributed variables, the Friedman test was used. The effect size was calculated with the use of Cohen’s D coefficient. All statistical analyses were conducted using GraphPad Prism 9 software (GraphPad Software, San Diego, CA, USA).

## 5. Conclusions

A decreasing trend in serum ferritin concentrations after Nordic walking training has been observed in patients with MM in remission taking part in this study, suggesting that the short duration of the training may not be sufficient to induce significant changes in ferritin concentrations.

Intracellular changes in mRNA levels of genes related to iron metabolism showed a tendency towards decreased expression after Nordic walking training cycles, indicating an influence of the training on these genes.

In conclusion, the results presented in this study suggest that the applied type of training may have beneficial effects in patients with MM. The moderate intensity exercise intervention applied in our study did not cause an increase in oxidative damage of macromolecules, which is a favorable effect. However, longer periods of training may be necessary to achieve significant favorable changes as observed in cited studies in healthy subjects. The present study provides further evidence that Nordic walking training is a safe form of physical activity for patients with MM in the disease remission phase, and as such should be recommended to patients. In our opinion, any treatment which induces decreases in genes and proteins associated with iron metabolism could be treated as positive and extend the period of remission, wherein a similar tendency refers to oxidative stress parameters. Thus, Nordic walking training is suitable for patients with MM; however, it is worth considering a longer period than 6 weeks of training.

## Figures and Tables

**Figure 1 ijms-24-15358-f001:**
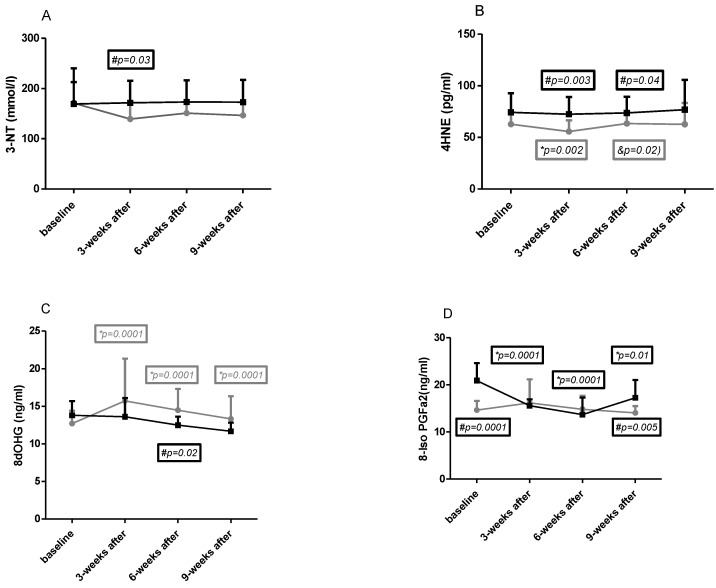
Changes in the concentrations of oxidative stress markers in the study groups ((**A**)—3-nitrothyrosine, (**B**)—trans-4-hydroxy-2-nonenal, (**C**)—8-OH-deoxyguanosine, (**D**)—8-iso-prostaglandin F2α). The results are presented as mean ± standard deviation (SD). NW (gray curve)—group undergoing a six-week cycle of Nordic walking training. CG (dark curve)—control group. *—statistically significant changes within each time point and baseline value. &—significant differences between 2–3. #—statistically significant differences between groups.

**Figure 2 ijms-24-15358-f002:**
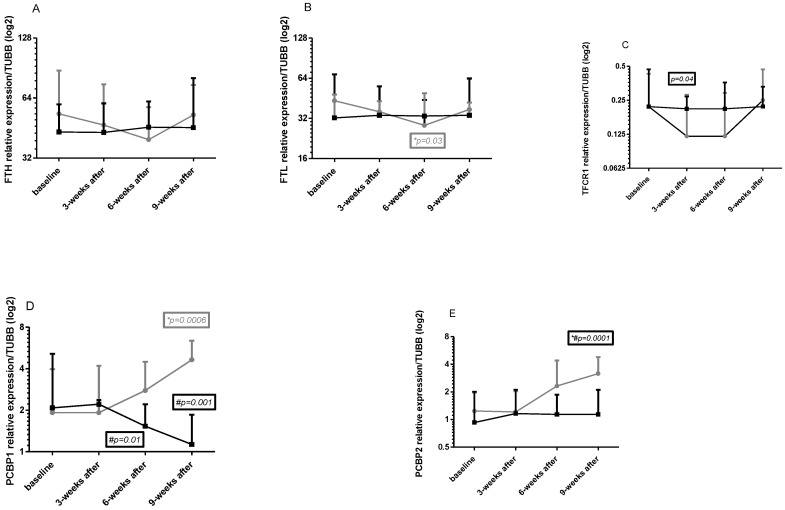
Changes in the relative gene expression in the study groups ((**A**)—ferritin heavy chain 1, (**B**)—ferritin light chain, (**C**)—transferrin receptor, (**D**)—poly(rC) binding protein 1, (**E**)—poly(rC) binding protein 2). The results are presented as mean ± standard deviation (SD). NW (gray curve)—group undergoing a 6-week cycle of Nordic walking training. CG (dark curve)—control group. *—statistically significant changes within the group compared to baseline. #—statistically significant differences between groups.

**Figure 3 ijms-24-15358-f003:**
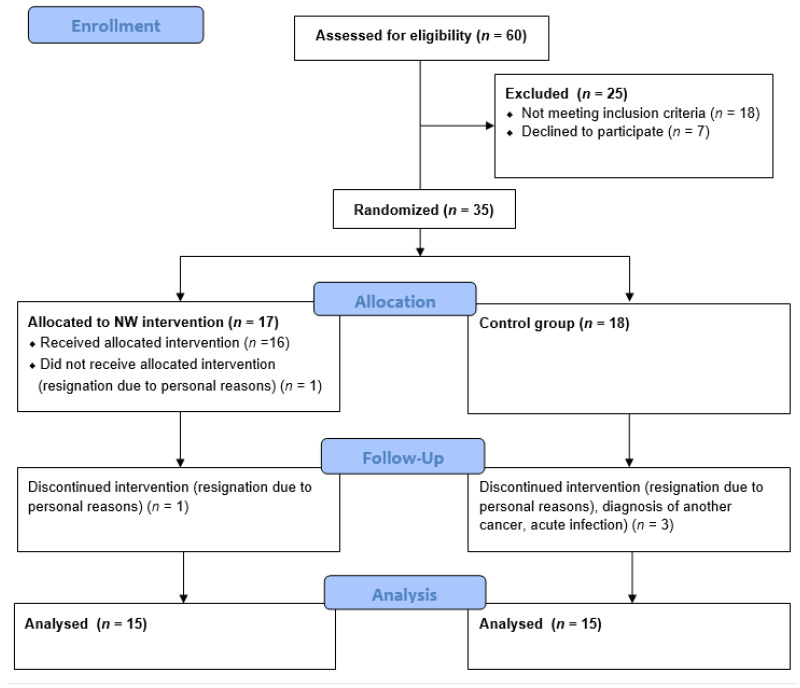
CONSORT 2010 patient flow diagram.

**Table 1 ijms-24-15358-t001:** Changes in the mean values of blood cells count in patients from both participating groups.

		Baseline	3 Weeks	6 Weeks	9 Weeks (Follow Up)	*p*
WBC [* 10^3^/μL]	NW	5.43 ± 1.23	5.80 ± 1.07	5.81 ± 1.31 *	5.37 ± 1.16 *	* 0.02
	CG	5.55 ± 0.86	5.54 ± 0.87	5.63 ± 0.86	5.53 ± 0.86	>0.05
	*p*	>0.05	>0.05	>0.05	>0.05	
	Cohen d	0.11	0.27	0.16	0.16	
RBC [* 10^6^/μL]	NW	4.28 ± 0.30 *^,&^	4.47 ± 0.29 *	4.33 ± 0.27 ^$^	4.47 ± 0.28 ^&,$^	* 0.01
	CG	4.23 ± 0.32	4.32 ± 0.34	4.27 ± 0.38	4.26 ± 0.41	^&^ 0.04
	*p*	>0.05	>0.05	>0.05	>0.05	^$^ 0.03
	Cohen d	0.24	0.63	0.18	0.60	
HGB [g/dL]	NW	13.16 ± 1.41	13.73 ± 1.31	13.42 ± 1.03	13.35 ± 1.01	>0.05
	CG	13.17 ± 0.82	13.18 ± 0.88	13.14 ± 0.88	13.12 ± 0.89	>0.05
	*p*	>0.05	>0.05	>0.05	>0.05	
	Cohen d	0.00	0.50	0.29	0.24	

Results are presented as mean ± standard deviation (SD). NW—group undergoing a six-week cycle of Nordic walking training. CG—control group. WBC—white blood cells count. RBC—red blood cells count. HGB—hemoglobin concentration. *^,&,$^—statistically significant changes within the group. Cohen d coefficient values interpretation: d (0.01) = very small, d (0.2) = small, d (0.5) = medium, d (0.8) = large, d (1.2) = very large, and d (2.0) = huge [19]. The values taken by the indicator d can be both below and above zero, with the absolute value being interpreted.

**Table 2 ijms-24-15358-t002:** Changes in the mean values of iron metabolism indicators in patients from both participating groups.

		Baseline	3 Weeks	6 Weeks	9 Weeks (Follow Up)	*p*
Iron [μmol/L]	NW	18.74 ± 6.38 *^,#^	17.79 ± 5.58 ^#^	14.85 ± 2.68 *	17.08 ± 9.22	* 0.029
	CG	14.06 ± 4.34 ^#^	14.36 ± 2.36 ^#^	16.52 ± 3.81	15.42 ± 2.36	>0.05
	*p*	0.026	0.036	>0.05	>0.05	
	Cohen d	0.86	0.80	−0.52	0.66	
Ferritin [ng/mL]	NW	92.62 ± 63.63 ^#^	93.90 ± 70.57 ^#^	95.48 ± 50.58	84.53 ± 62.50 ^#^	>0.05
	CG	143.19 ± 68.58 ^#^	146.20 ± 47.82 ^#^	122.04 ± 42.53	135.49 ± 49.18 ^#^	>0.05
	*p*	0.042	0.023	>0.05	0.018	
	Cohen d	−0.76	−0.87	−0.57	−0.90	
Transferin [g/L]	NW	2.41 ± 0.42 ^#^	2.35 ± 0.34	2.37 ± 0.37 *	2.36 ± 0.32	>0.05
	CG	2.15 ± 0.25 ^#^	2.15 ± 0.24	2.28 ± 0.21	2.21 ± 0.23	>0.05
	*p*	0.049	>0.05	>0.05	>0.05	
	Cohen d	0.75	0.68	0.30	0.53	
TfS [%]	NW	40.1 ± 14.4 *	38.6 ± 12.5	31.9 ± 7.3 *	40.1 ± 20.6	* 0.037
	CG	32.4 ± 8.2	33.3 ± 3.0	36.5 ± 8.5	35.1 ± 5.3	>0.05
	*p*	>0.05	>0.05	>0.05	>0.05	
	Cohen d	0.66	0.36	−0.58	3.05	

Results are presented as mean ± standard deviation (SD). NW—group undergoing a six-week cycle of Nordic walking training. CG—control group. TfS—transferrin saturation. *—statistically significant changes within the group. ^#^—statistically significant differences between groups. Cohen d coefficient values interpretation: d (0.01) = very small, d (0.2) = small, d (0.5) = medium, d (0.8) = large, d (1.2) = very large, and d (2.0) = huge [19]. The values taken by the indicator d can be both below and above zero, with the absolute value being interpreted.

**Table 3 ijms-24-15358-t003:** Inclusion and exclusion criteria.

Inclusion Criteria:	Exclusion Criteria:
- Multiple myeloma in the plateau stage without cytostatic treatment - Bisphosphonate therapy - Overall good patient condition (ECOG scale: 0.1.2)	- Significant liver and kidney damage (evaluated during qualification to the project) - Acute respiratory infection or other infectious disease - Another malignancy - Recent fall from own height resulting in injury

ECOG—Eastern Cooperative Oncology Group.

## Data Availability

Data are available on request from the corresponding author.

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
