# Peer review of "The Effect of a Six-Week Nordic Walking Training Cycle on Oxidative Damage of Macromolecules and Iron Metabolism in Older Patients with Multiple Myeloma in Remission—Randomized Clinical Trial"

_ijms, 2023, doi:10.3390/ijms242015358_

Round 1
Reviewer 1 Report
Appointments:
Congratulations for this excellent work. These data should be published to encourage further studies in this line of research. However, some considerations about this manuscript need to be revised.
The term control group was expressed as GK, CG and CON; please choose CG for all in-text citations.
The first figure should be that of the CONSORT 2010 Patient Flow Diagram;
The first table must contain the clinical data of the two groups. This would avoid having to describe the data in the text of the methods or results. In addition, the statistical analysis information must be described in this first table. The main idea is to show all clinically relevant features that need to be homogeneous across groups. Among the main variables are the values ​​of iron, ferritin, transferrin, and transferrin saturation. These variables need to be homogeneous at baseline. A statistical analysis must be clearly presented for the initial data. Complementary information regarding the clinical diagnosis of patients should also be informed in table 1. For example, diagnosis time, treatment cycles performed, pharmacological treatment, patients' clinical phase (induction, remission, consolidation), among other issues that could describe the physical conditions in general. After introduction, is essential to insert information about the methods. This should be the first step in describing a clinical trial method. Follow the guidelines outlined in CONSORT 2010 for rearranging the order of information. Introduction, method, results, discussion, and conclusions.
The presentation of data in graphic figures needs to make clear in the captions the exact moment of data collection. There were 6 weeks of intervention and 4 moments of comparison appear in the graphs. This discrepancy can confuse the reader if the information is not clearly described.
I ask authors to standardize the y-axis scale of the graphs, always starting at zero.
I also request that information regarding non-significant statistical comparisons be presented in an unbiased manner. Avoid non-significantly increased or reduced. Give preference to neutral terms, as they oscillate over time. Alternatively, it can be said that in certain aspects there is an increasing or decreasing trend.
In the discussion, line 228. Acute myeloid leukemia (AML) does not seem to be a recurring term that needs abbreviation. On the other hand, multiple myeloma appears several times in the text in an unabbreviated form.
Throughout the discussion, the temporal description of events related to the intervention, evaluation and follow-up can be improved. The temporal sequence of events needs to be addressed in the text in a more user-friendly way. For example: 3 weeks after the end of intervention could be described as follow-up (phase IV).
In the methods section: I suggest that Table 2 be replaced by a new Table 1, where all relevant clinical data are described at the baseline of the study, comparing the data of the two groups. See the suggestions described above.
In conclusion of the study: I ​​suggest adding that... NW training is a safe form of physical activity for patients with MM (in the disease remission phase) ...
Author Response
Dear Reviewer,
We would like to thank you for taking the time to review our manuscript and for your favorable opinion of it. Your valuable remarks helped to improve the scientific quality of our manuscript. Please find our responses to your comments in the attached file.

Reviewer 2 Report
In the introduction, please use abreviations throughout all the manuscript once explained (e.g. MM, line 55 page 2). Same for nordik walking, please explain the abbreviation before using it (line 90, page 2). Revise this not only in the introduction, but also for the methods, results and discussion sections of the manuscript.
In the results section, table 1 does not have a proper caption. Please revise this on all tables and include information related to the data on the tables. However, results are well explained and figures are easy to understand.
Regarding the discussion, this section is well structured and the results of this manuscript are well compared with similar studies.
However, in the methods section, it is not indicated if you performed a sample size calculation or did you just take a sample size of convenience. Please include this information if possible, as well as the information about the randomization proccess. Where the CG participants asked if they wanted to participate on the NW group after finishing the study? Did the CG receive any indications of performing exercise?
I would reccomend the authors to change this in order to improve their manuscript so that it can be published.
Author Response
Dear Reviewer,
We would like to thank you for taking the time to review our manuscript. Your insightful remarks helped to improve the scientific quality of our manuscript. Please find our responses to your comments in the attached file.

Reviewer 3 Report
According to the authors, the main objective of this study was to demonstrate the alterations in several molecules associated with iron metabolism, both by its evaluation in serum and by gene expression on leukocytes, and also the oxidative damage in patients with multiple myeloma submitted to an exercise training program composed of 6-week cycle of Nordic walking training cycle. Even though the purpose and some interesting data were presented, there are another couple of factors that should be concerned.
Please revise the title, since it is difficult to understand the meaning of the study.
Title suggested: "The effect of a 6-week Nordic walking training cycle on oxidative damage of macromolecules and iron metabolism in older patients with multiple myeloma"
In the Abstract section:
1) Please remove the word "cycle" in the objective since this word is presented twice.
2) Please revise and unify the description of the parameters associated with iron metabolism assessed in the study.
3) The description of results is very poor, and, in this way, it is impossible to identify the relevance of the information provided. For instance: What were the p-values? What gene(s) was (were) altered? What time the "alterations" were found? Do the "differences" found were between the volunteer groups or among the study's time points assessed?
4) Since the description of the results is insufficient to support any conclusion, it is difficult to evaluate this topic.
In the Introduction section:
5) The authors mentioned, in lines 64-66, that "However, numerous studies have confirmed that moderate-intensity physical exercise does not lead to an increase in the production of free radicals [6,7]..." However, I suggest revising this sentence, since only two articles are few to support this information. In addition, in the same reference [6], it was mentioned that "However, recent reports suggested that even low or moderate intensity can induce oxidative stress, suggesting that exercise volume (duration × intensities) and failed antioxidant defense system are the primary mediators of exercise-induced oxidative stress."
6) In the description of the study´s objectives (lines 101-102), the authors mentioned that oxidative stress was assessed. Based on the definition of oxidative stress, in which "it is a phenomenon caused by an imbalance between the production and accumulation of oxygen-reactive species (ROS) in cells and tissues and the ability of a biological system to detoxify these reactive products by antioxidant agents", in the present study the authors did not assess antioxidant agents. Thus, it is impossible to respond to this aim, since one side of oxidative stress was not evaluated.
In the Results section
7) Line 108, what means GK?
8) Why was the leukogram not assessed?
9) I recommend presenting the data of the first two sentences in a table, including the p-value. In addition, it is important to mention the absolute number, as well as the ratio, of the men and women who participated in the study.
10) Since iron metabolism was assessed in the study, why the authors did not evaluate the hemoglobin at all time points?
11) Please revise the Table 1 legend, and also add, at least, the significant p-values.
12) In line 134, the authors mentioned that "...no correlation between iron and ferritin was found in both groups." Did was performed a statistical test for correlation analysis?
13) Since the results are presented in different ways, it is very difficult to understand their meaning and relevance. For instance, in some descriptions are presented p-values, in another the Cohen d values, and in others none of them. In addition, it is impossible to observe any values the Figure 2.
14) I recommend separating the description of results related to the intragroup analysis from the other results regarding the intergroup analysis.
In the Discussion and Conclusion sections:
15) Since it is necessary to revise and rewrite the results, it is difficult to evaluate both the "Discussion" and "Conclusion" sections.
In the Material and Methods section:
16) How was perform the randomization of volunteers?
17) In lines 430-431, the authors declared that "To verify correlation between plasma iron or ferritin and genes expression values, Spearman's rank correlation coefficient was calculated." However, these results were not presented.
Moderate editing of English language required
Author Response

(The authors gave the same response as above.)

Reviewer 4 Report
Dear author the paper is interesting. There are several errors that need to be addressed before publication.
1) In the title has to be specified that MM patients were in remission, and that this is a randomized trial
2) The abstract is about 350 words, normally the Journal consent only 200 words, please reduce the length of the abstract
3) Methods section NEED to be before result section.
4) Inclusion criteria need to be better clarify: “Multiple myeloma in the plateau stage without cytostatic treatment” Last treatment? Actual monoclonal component? What do you mean of measured to define this item
5) “Permissible bisphosphonate therapy” this isn’t an inclusion criterion. Specify that this treatment is allowed and performed in all the patients
6) “Overall good patient condition” specify…
7) Exclusion criteria need to be better clarify: “Significant liver and kidney damage” specify…
8) Which medicaments were allowed during this trial? And which drugs take the patients?
9) Which chemotherapy followed the 30 patients enrolled before the treatment?
10) Image need to be improved, please use different color or different way the line (dotted line) in this way are difficult to understand the two different groups
11) P value are shown in figure number 2, but not in figure number 1. please uniform.
12) P value are shown in paragraph “Macromolecules oxidative damage parameters” but not in “Parameters of iron metabolism in serum” and “Gene expression in leukocytes” please uniform.
13) Could you add data on Hemoglobin both in table 1, figure 1 and in the text at the 4 check point time
14) Paragraph “Iron metabolism in multiple myeloma” in the discussion need to be shorted isn’t the core of your paper.
15) Page 8 line 314-322 are similar to your conclusion, please integrate those paragraphs in one conclusion
16) Quote number 1, 2, 9, 17 are in polish, and look like a self-citation that isn’t necessary. Please remove it and add a paper in English like international guideline on MM
Author Response

(The authors gave the same response as above.)

Round 2
Reviewer 2 Report
I thank the authors for addressing all comments in my review and other reviewers. In my opinion, this manuscript is ready to be published
Author Response
Dear Reviewer,
We would like to thank for your time and assistance in improving the quality of our manuscript.
Reviewer 3 Report
I would like to congratulate the authors for revising the manuscript, which significantly improved its meaning.
No comments.
Author Response
Dear Reviewer,
We would like to thank you for your time and assistance in improving the quality of our manuscript.
Reviewer 4 Report
Dear authors, the article is improved and can be published in the revised version.
It needs only a small modification:
- Figure 1 has two different presentations, unify it with the second type. Also, the eight figures have the same subtitle, repeated twice A,B, C, D. It is better to continue with the alphabet A,B, C, D, E, F, G, H.
Author Response
Dear Reviewer,
Figures have been corrected as requested.
We also would like to thank for your time and assistance in improving the quality of our manuscript.